

# A multi-class classification algorithm based on hematoxylin-eosin staining for neoadjuvant therapy in rectal cancer: a retrospective study

Yihan Wu[1,2,*], Xiaohua Liu[3,*], Fang Liu[4], Yi Li[2,3], Xiaomin Xiong[2,3], Hao Sun[5], Bo Lin[2], Yu Li[4] and Bo Xu[1,2]

[1] School of Medicine, Chongqing University, Chongqing, China
[2] Chongqing Key Laboratory of Intelligent Oncology for Breast Cancer, Chongqing University Cancer Hospital, Chongqing, China
[3] Bioengineering College of Chongqing University, Chongqing, China
[4] Department of Pathology, Chongqing University Cancer Hospital, School of Medicine, Chongqing University, Chongqing, China
[5] Department of Gastrointestinal Cancer Center, Chongqing University Cancer Hospital, Chongqing, China
[*] These authors contributed equally to this work.

Corresponding authors
Yu Li, liyu100@cqu.edu.cn
Bo Xu, xubo731@cqu.edu.cn

## ABSTRACT

Neoadjuvant therapy (NAT) is a major treatment option for locally advanced rectal cancer. With recent advancement of machine/deep learning algorithms, predicting the treatment response of NAT has become possible using radiological and/or pathological images. However, programs reported thus far are limited to binary classifications, and they can only distinguish the pathological complete response (pCR). In the clinical setting, the pathological NAT responses are classified as four classes: (TRG0-3), with 0 as pCR, 1 as moderate response, 2 as minimal response and 3 as poor response. Therefore, the actual clinical need for risk stratification remains unmet. By using ResNet (Residual Neural Network), we developed a multi-class classifier based on Hematoxylin-Eosin (HE) images to divide the response to three groups (TRG0, TRG1/2, and TRG3). Overall, the model achieved the AUC 0.97 at 40× magnification and AUC 0.89 at 10× magnification. For TRG0, the model under 40× magnification achieved a precision of 0.67, a sensitivity of 0.67, and a specificity of 0.95. For TRG1/2, a precision of 0.92, a sensitivity of 0.86, and a specificity of 0.89 were achieved. For TRG3, the model obtained a precision of 0.71, a sensitivity of 0.83, and a specificity of 0.88. To find the relationship between the treatment response and pathological images, we constructed a visual heat map of tiles using Class Activation Mapping (CAM). Notably, we found that tumor nuclei and tumor-infiltrating lymphocytes appeared to be potential features of the algorithm. Taken together, this multi-class classifier represents the first of its kind to predict different NAT responses in rectal cancer.

## INTRODUCTION

Cancers originated from colon and rectal are the third most commonly diagnosed and the second deadliest worldwide (*Sung et al., 2021*). Rectal cancer is distinctive to colon cancer, owing to its anatomical location, etiological factors and environmental associations. For locally advanced rectal cancer, neoadjuvant therapy (NAT) is a standard preoperative treatment strategy, which is conducive to the preservation of the anal sphincter muscle, to the complete surgical removal of tumor cells, as well as to the reduction of tumor micrometastasis and recurrence (*Benson et al., 2020*). The efficacy of NAT is typically evaluated by the postoperative pathological assessment, reported by the tumor regression grade (TRG) (*Karagkounis et al., 2019*; *National Health Commission of the People's Republic of China, 2020*). The American Joint Committee on Cancer (AJCC) TRG system (*Kim et al., 2016*) consists of Grades 0, 1, 2, and 3, among which TRG0 is regarded as pathological complete response (pCR), indicating that there are no residual active tumor cells in the primary nidus. TRG1 indicates a moderate response with single or small groups of tumor cells remaining. TRG2 reflects a minimal response with residual cancer outgrown by fibrosis. TRG3 is regarded as a poor response with large areas of residual cancer in the primary lesion, and few or no tumor cells are killed. It is believed that TRG grading has predictive values as there are significant correlations between TRGs and the clinical outcome in the disease (*Belluco et al., 2011*; *de Campos-Lobato et al., 2011*). Unfortunately, there are less than 30% of patients achieving pCR (*Monson & Arsalanizadeh, 2017*). For patients who cannot achieve it, especially for TRG2 and 3, NAT is not beneficial as it might weaken their immune system and delay other treatment. Therefore, approaches to NAT risk stratification are urgently needed.

Over the years, significant efforts have been invested in identifying risk factors of rectal cancer NAT. On the biological side, several categories of biomarker identifications have been explored. For example, studies have shown that the pretreatment tumor marker CEA level (*Wallin et al., 2013*; *Das et al., 2007*), HEP2, GELS, and S10A8 protein expression in serum (*Wang et al., 2022*), the number of tumor-infiltrating lymphocytes (TILs) (*Matsutani et al., 2018*), subsets of tumor-infiltrating and peripheral blood immune cells (*Zhu et al., 2022*), inflammatory cancer-associated fibroblasts (iCAFs) (*Nicolas et al., 2022*), and the CD44 mRNA level (*Huh, Lee & Kim, 2014*), can be used as biomarkers to predict treatment response in rectal cancer NAT. Higher TILs, the CD4+/CD8+ T cell ratio, HEP2 and GELS, and lower iCAFs, S10A8 and pretreatment CEA levels are associated with better responses.

In recent years, artificial intelligence (AI) based programs have been explored in identifying features of rectal cancer NAT. Based on traditional machine learning (ML) and deep learning (DL), AI can discover and extract hidden information in images. Therefore it plays an increasingly significant role in various aspects of tumor screening, diagnosis, assessment of efficacy, and prediction of prognosis (*Coudray et al., 2018*; *Bilal et al., 2021*; *Zhou et al., 2020*; *She et al., 2020*; *Courtiol et al., 2019*; *Lin et al., 2022*). Among them, studies have used radiological images involving CT (*Kurata et al., 2021*) and MRI (*Liu et al., 2017*) to predict the treatment response to NAT in rectal cancer. In addition to radiological images, pathological images have also been used to extract

features that they might be associated with the therapeutic response. Pathological images contain macroscopic histological features and microscopic cellular information, and these characteristics intuitively provide the essential role of the occurrence and development of the disease. DL can extract many high-dimensional features through various algorithms, which are invisible to human eyes. For example, some algorithms can predict microsatellite status (*Yamashita et al., 2021*) and prognosis (*Kather et al., 2019*) based on pathological images of rectal cancer. In addition, studies (*Zhang et al., 2020*; *Shao et al., 2020*) have utilized ML methods to manually extract features from pathological images of rectal cancer patients to predict the pCR.

Despite these promising results, it is noted that all these studies are aimed at identifying features or biomarkers for the pCR patients. Owing to the multi-category grading system, it is apparent that solely finding pCR patients is not sufficient. As a matter of fact, identifying TRG2 and TRG3 patients, which are less responsive to NAT, might be more important, as NAT do not benefit these patients. Although there is a recent study (*Jepsen et al., 2022*) that has developed a multi-class quantitative evaluation algorithm for pathological images of rectal cancer after neoadjuvant therapy, multi-class prediction algorithms have not been reported.

In the present study, we established a three-class classification method to predict the efficacy of NAT in patients with rectal cancer. We show a robust performance in categorizing TRG01, TRG1/2 and TRG3. In addition, we established a heat map based on the CAM method to analyze the interpretability of pathological images in different treatment responses.

## MATERIALS & METHODS

### Patient selection
A total of 94 patients who received NAT in Chongqing University Cancer Hospital (CUCH) from 2011 to 2022 were retrospectively enrolled. The criteria for inclusion comprised: (1) Diagnosis of primary rectal cancer through pathological biopsy before NAT; (2) receiving of the NAT regimen based on Capecitabine or 5-FU combined with Oxaliplatin; (3) availability of preoperative pathological biopsy sections; (4) have underwent surgery after NAT and have TRG evaluation results. The criteria for exclusion comprised: (1) Other treatments before NAT; (2) lack of preoperative pathological biopsy section; (3) poor quality of H&E-stained images. This study was approved by the Ethics Committee of Chongqing University Cancer Hospital (Ethics number: CZLS2022185-A).

### Pathological assessment of response and data collection
Specimens were prepared onto glass slides and stained with hematoxylin & eosin (H&E). TRG grades were evaluated by two experienced pathologists, and samples with discordant scores were reevaluated until an agreement was reached. In addition, clinical data including the age at diagnosis, gender, tumor/node(T/N) stage, CEA and CA199 levels before treatment, and treatment regime were collected. Due to the institutional policy, the pathological image data are not publicly available.

## Image preprocessing

H&E-stained slides were produced at 40× magnification by the KFBio KF-PRO-005-HI digital scanner. There were 133 slides collected. Whole slide images (WSIs) were annotated using QuPath open-source software (version 0.3.2). The annotation of tumor tissues in the WSIs was performed by two pathologists. One pathologist was responsible for delineating, and the other for examining. To reduce the computing consumption, the WSIs and their corresponding annotations were split into overlapping 512 × 512 pixels tiles at 0.238 μm/pixel resolution (equivalent to 40× magnification) and 1 μm/pixel resolution (equivalent to 10× magnification) with overlapping segmentation, which stride was 0.5. Tiles with tissues less than 50% were removed manually.

All tiles were color normalized with the structure-preserving color normalization (SPCN) method (*Vahadane et al., 2016*). Different types of data augmentation were applied, including random vertical flip (0.5), random horizontal flip (0.5), and brightness/contrast/saturation/hue changes (1).

## Construction and visualization of response classifier

The WSIs from CUCH were split at a ratio of approximately 4:1, with 113 WSIs in the training set and 20 WSIs in the test set. The training set consisted of 166,919 tumor ROI tiles at 40× magnification. There were 14 slides (27,598 tiles) from TRG0, 70 slides from TRG1/2 (98,102 tiles), and 28 slides (41,219 tiles) from TRG3. We included multiple slides per patient, a patient had one or two slides. After a series of data augmentation techniques, the augmented training set increased to three times the original training set. The test set was seven slides (15,506 tiles) from TRG0, seven slide (13,407 tiles) from TRG1/2, and six slides (10,263 tiles) from TRG3. For feature extracting, we utilized a deep residual network (Resnet101) without the final softmax classification layer and average pooling layer, pretrained with ImageNet. After the 2048-dimension feature vector were extracted, multiple full connected layers were used to non-linear mapping. At the end, a classification layer was added. In the training process, the tiles with 512 × 512 pixels were re-sized to 256 × 256 pixels and then input into the network. Among them, the patient's treatment responses were considered as the target label. We optimized the response classifier using a five-fold cross-validation approach, which involves randomly dividing the training set into five balanced subsets. Based on TRG classes, each fold was divided into 4:1 from different TRG class to extract training samples and validation samples. The hyperparameters include a batch size of 32, an initial learning rate of 0.001, and total epochs of 20 (decreasing learning rate by 0.1 every five epochs). In the post-processing stage, the average score of the tumor tiles of each WSI was taken as the prediction probability of the WSIs, which was regarded as the criterion to judge the efficacy of NAT. The response classifier structure was implemented in Python, using Pytorch backends (Python 3.9.7, Pytorch 1.11.0, SKLeran 0.24.2). We used the same network structure, batch size, and learning rate to train and verify the model at 10× magnification. The initialized parameters with pre-training weights in ImageNet.

To analyze features used by the classifier to make its decision, we performed Smooth Grad-CAM (class activation mapping) (*Omeiza et al., 2019*) to identify regions within each

tile that the neural network uses to generate predictions. Then, we used cell detection and classification functions in Qupath to identify three types of cells: tumor cells, tumor-infiltrating lymphocytes (TILs), and stroma (*Acs et al., 2019*). TILs% calculated as follows:

$$TILs\ \% = \frac{TILs}{TILs + Tumor\ cells} \times 100\%.$$

## Statistical analysis

We evaluated models at the tile and slide levels. The neural network yielded a probability value for every class of interest in each tile. Then we averaged the probabilities of each tile from the patients' slides to assign a final probability to each patient. The probabilities calculated by the neural network were used to generate the area under the curve (AUC) for the receiver operating characteristic (ROC) curve. Using macro-AUC and micro-AUC as metrics, we compared the performance of the models at different magnifications. We also used Precision, Sensitivity, Specificity, and the multi-class Matthews' correlation coefficients (MCC) to assess the performance of models. The statistical analyses were performed using SKLeran 0.24.2 in Python (scikit-learn.).

# RESULT

## Workflow for predicting NAT response with a multi-class classification algorithm

A total of 94 patients were included in this study based on the inclusion/ exclusion criteria. Table 1 summarizes the characteristics of TRGs of the cohort. The workflow of our studies is shown in Fig. 1. Firstly, we collected and preprocessed the HE images from the patients, including scanning, delineating and segmenting the ROIs. In this step, a total of 166, 919 tiles were obtained. Then we normalized the color of tiles by the SPCN (*Vahadane et al., 2016*). Secondly, we input the tiles into the neural network in the form of tiles at $10\times$ and $40\times$ magnification, and extract the features through ResNet-101 to deduce the probabilities and predict the treatment responses: TRG0, TRG1/2, or TRG3. Finally, we constructed a visual heat map of tiles using CAM to explore the medical interpretability of the model.

## Model performance evaluation under multi-class classification tasks on different scale

To test the model performance on different fields of view, HE tiles were preprocessed with two magnifications, including that of $40\times$ and $10\times$. We first developed the response classifier at $40\times$ magnification. At the tile level, the macro-AUC of the three-class classification model was 0.82, and the micro-AUC was 0.86 on the cross-validation of the training set, and the macro-AUC of 0.72 and the micro-AUC of 0.72 on the test set, as shown in Figs. 2A and 2E. The model revealed a stable performance with the macro-AUC of 0.97 and the micro-AUC of 0.96 at the slide level, as shown in Fig. 2B. In order to compare the performance on different scales, with this dataset we trained and tested the classifier at $10\times$ magnification. The AUCs of the tile and slide levels are shown in Figs. 2C and 2D. The macro-AUC and micro-AUC were 0.75, 0.78 and 0.79, 0.86, respectively. And the result

**Table 1** Characteristics of the patients.

| characteristics | TRG0 | TRG1/2 | TRG3 | *p* value |
|---|---|---|---|---|
| Gender (%) | | | | 0.779 |
|    Male | 12(63.2%) | 32(62.7%) | 17(70.8%) | |
|    Female | 7(36.8%) | 19(37.3%) | 7(29.2%) | |
| Age, mean ± SD | 52.74 ± 10.76 | 56.65 ± 11.36 | 56.50 ± 13.23 | 0.443 |
| Tumor location | | | | 0.205 |
|    Upper | 2(15.4%) | 3(6.1%) | 4(18.2%) | |
|    Middle | 4(30.8%) | 20(40.8%) | 16(54.5%) | |
|    Lower | 7(53.8%) | 26(53.1%) | 6(27.3%) | |
| Pretreatment T stage (%) | | | | 0.083 |
|    T1 | 0(0%) | 0(0%) | 0(0%) | |
|    T2 | 0(0%) | 0(0%) | 2(8.7%) | |
|    T3 | 4(21.1%) | 19(38.8%) | 6(26.1%) | |
|    T4 | 15(78.9%) | 30(61.2%) | 15(65.2%) | |
| Pretreatment N stage (%) | | | | 0.328 |
|    N0 | 1(5.3%) | 3(6.0%) | 4(16.7%) | |
|    N1-N2 | 13(68.4%) | 38(76.0%) | 13(54.2%) | |
|    Nx | 5(26.3%) | 9(18.0%) | 7(29.2%) | |
| Pretreatment CEA | | | | 0.448 |
|    ≤5 ng ml | 13(72.2%) | 30(58.8%) | 17(70.8%) | |
|    >5 ng ml | 5(27.8%) | 21(41.2%) | 7(29.2%) | |
| Pretreatment CA199 | | | | 0.879 |
|    ≤34 U/ml | 14(77.8%) | 37(74.0%) | 17(70.8%) | |
|    >34 U/ml | 4(22.2%) | 13(26.0%) | 7(29.2%) | |
| Neoadjuvant chemotherapy regime | | | | 0.856 |
| Based on Capecitabine | 16 (84.2%) | 39 (76.5%) | 19 (79.2%) | |
| Based on 5-FU | 3 (15.8%) | 11 (21.6%) | 4 (16.7%) | |
| Cycles | 4.39 ± 1.82 | 3.55 ± 2.27 | 3.48 ± 2.87 | 0.383 |
| Underwent neoadjuvant radiotherapy | | | | 0.009 |
| Yes | 18 (94.7%) | 40 (78.4%) | 12 (50.0%) | |
| No | 1 (5.3%) | 11 (21.6%) | 11(45.8%) | |

**Notes.**

CEA, carcinoembryonic antigen; CA199, carbohydrate antigen 19-9; SD, standard deviation.

of the test set and the variability on the 5-folds cross validation are shown in Figs. 2E–2H, Tables 2, and 3.

After comparing the overall performance of models, we determined the specific classification performance of the model for each group shown in Fig. 3. Specifically, the performance of the model at the 40× magnification is better than at the 10× magnification, which indicates that the local cellular features extracted from the model were more conducive to the differentiation of curative effects than the global tissue features. The detailed AUCs under tile and slide level are provided in Table 4.
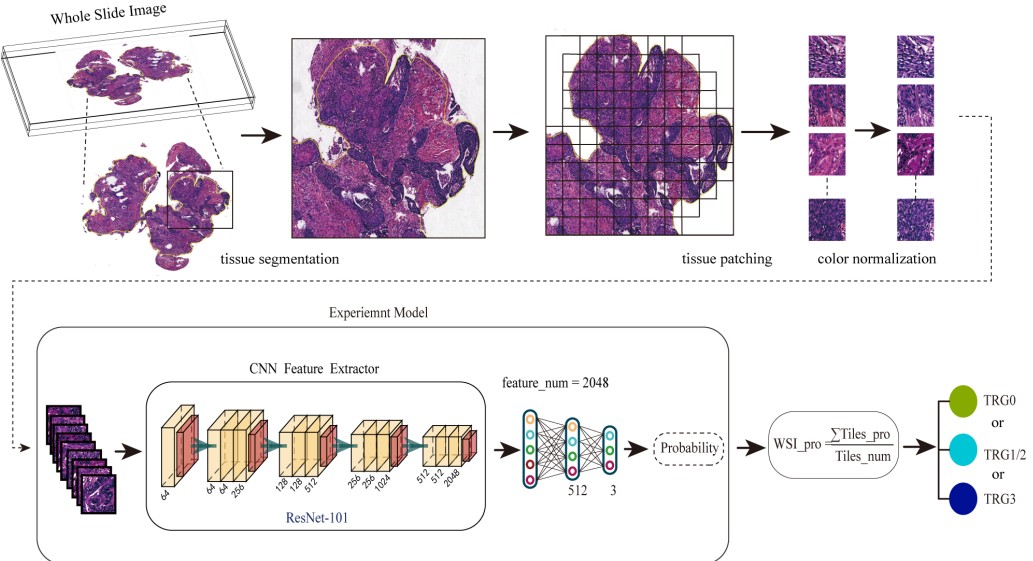

**Figure 1 The workflow for predicting the NAT response using the multi-class classification algorithm.** Firstly, we preprocessed the HE images, including scanning the images, delineating and segmenting the tumor tissues, and normalizing the color of tiles. Secondly, we input pathological images into the neural network in the form of tiles to deduce the probabilities. Finally, the WSIs were divided into different treatment responses by predicting probabilities.

## Medical interpretability analysis

To explore the relationship between histopathological features and the efficacy of NAT, we developed a post-hoc explanation of a trained model at the 40× and 10× magnifications, based on the CAM algorithm. The model generated a heat map for each tile, and the areas with large contribution from the model are displayed in red. As shown in Figs. 4A and 4B, overlaying the original tile images with the image of the CAM analyses revealed that tumor tissues including tumor cells and TILs seem to contribute the most in our model, while stroma contributes less to this model. Among them, TILs play an important role in the decision to classify responses, either with 40× or 10× magnifications. Specifically, tiles predicted with TRG0 appeared to be denser in the number of TILs than that with TRG1/2 or TRG3. As shown in Fig. 4E, it was observed that the proportion of TILs was significantly associated with the treatment response. The median TIL % in the TRG0 group was significantly higher than that in the other two groups (the TRG0 group was 35.1%, the TRG1/2 group 24.7%, and TRG3 group 16.9%).

## Correlation analysis of TRG prediction

As shown in Table 5, at the 40× magnification, the MCC in the prediction of TRG0,1/2 and 3 at the tile level were 0.499, 0.501 and 0.425, respectively, while the MCC in the prediction of TRG0, 1/2 and 3 at the slide level were 0.617, 0.734, and 0.683, respectively. At the 10× magnification, the MCC in the prediction of TRG0,1/2 and 3 at the tile level were 0.623, 0.233 and 0.221, respectively, while the MCC in the prediction of TRG0, 1/2 and 3 at the slide level were 0.690, 0.316, and 0.298, respectively. According to the MCC

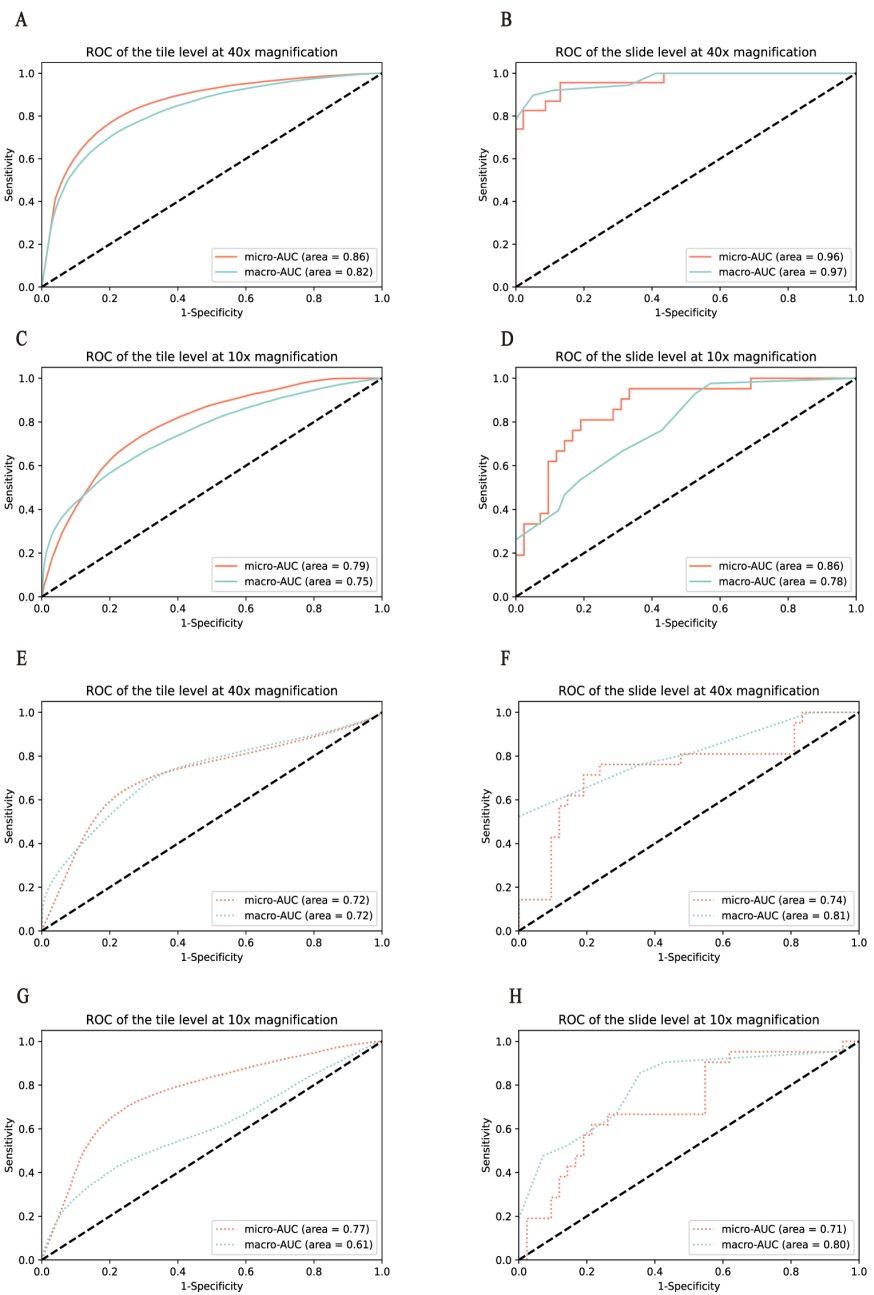

**Figure 2 Performance of the model was measured in terms of the AUC of the ROC curves.** Models were validated and tested on two levels. (A–D) represent the results of the model in the training set, and (E–H) represent the results of the model in the test set. Among them, (A ,C, E, and G) represent the performance comparison between 40× and 10× magnification of the model at the tile level. (B, D, F, and H) represent the performance comparison between 40× and 10× magnification of the model at the slide level.

results, TRG1/2 and TRG3 were weakly differentiated at low magnification. We also found that the proportion of TILs in TRG1/2 and TRG3 was lower than that in TRG0, and this microstructure could not be clearly displayed at low magnification if the proportion was

**Table 2  Macro-AUC and Micro-AUC for the training set and test set at 40× magnification.**

|  | Slide level | | Tile level | |
|---|---|---|---|---|
|  | **Macro-AUC** | **Micro-AUC** | **Macro-AUC** | **Micro-AUC** |
| Validation 1 | 0.97 | 0.96 | 0.82 | 0.86 |
| Validation 2 | 0.76 | 0.82 | 0.57 | 0.67 |
| Validation 3 | 0.78 | 0.84 | 0.68 | 0.77 |
| Validation 4 | 0.90 | 0.93 | 0.70 | 0.75 |
| Validation 5 | 0.94 | 0.94 | 0.93 | 0.94 |
| Test | 0.81 | 0.74 | 0.72 | 0.72 |

**Table 3  Macro-AUC and Micro-AUC for the training set and test set at 10× magnification.**

|  | Slide level | | Tile level | |
|---|---|---|---|---|
|  | **Macro-AUC** | **Micro-AUC** | **Macro-AUC** | **Micro-AUC** |
| Validation 1 | 0.80 | 0.87 | 0.71 | 0.77 |
| Validation 2 | 0.61 | 0.74 | 0.68 | 0.82 |
| Validation 3 | 0.73 | 0.81 | 0.65 | 0.77 |
| Validation 4 | 0.89 | 0.87 | 0.64 | 0.76 |
| Validation 5 | 0.78 | 0.86 | 0.75 | 0.79 |
| Test | 0.80 | 0.71 | 0.61 | 0.77 |

low. Therefore, this may be the reason for the low MCC in pathological images with low magnification. Meanwhile, according to the MCC analysis results, we found that TRG0 had better predictability at low magnification, while TRG1/2 and TRG3 hah relatively better predictability at high magnification. Therefore, combination of low magnification and high magnification pathological images to construct a multi-scale joint prediction model might improve the accuracy and robustness of TRG classification prediction for NAT.

## DISSCUSION

For locally advanced rectal cancer, NAT has become a standard treatment method that can improve the quality of life and overall survival. Despite the benefit of the treatment modality, about 40% of patients are not benefited from it. For this reason, identification of biomarkers (biological or multiomics signatures) or image markers (from radiological and/or pathological image features) for NAT responses is of clinical significance. Risk stratification for NAT represents the opportunity to change the chemo-radiotherapy regimens or switch to other treatments as early as possible. However, most studies toward this goal are limited to the prediction power to identifying pCR patients. Here we present a new multi-class DL algorithm that can identify three categories of the NAT responses: very sensitive (TRG0), moderately sensitive (TRG1/2) or resistant (TRG3). The performance of the algorithm is demonstrated in the two magnifications.

Over the last few years, various ML and DL algorithms have been applied to pathological image classification tasks, which allow disease identification and risk stratification. Studies have shown that pathological images can provide sub-visual information about molecular

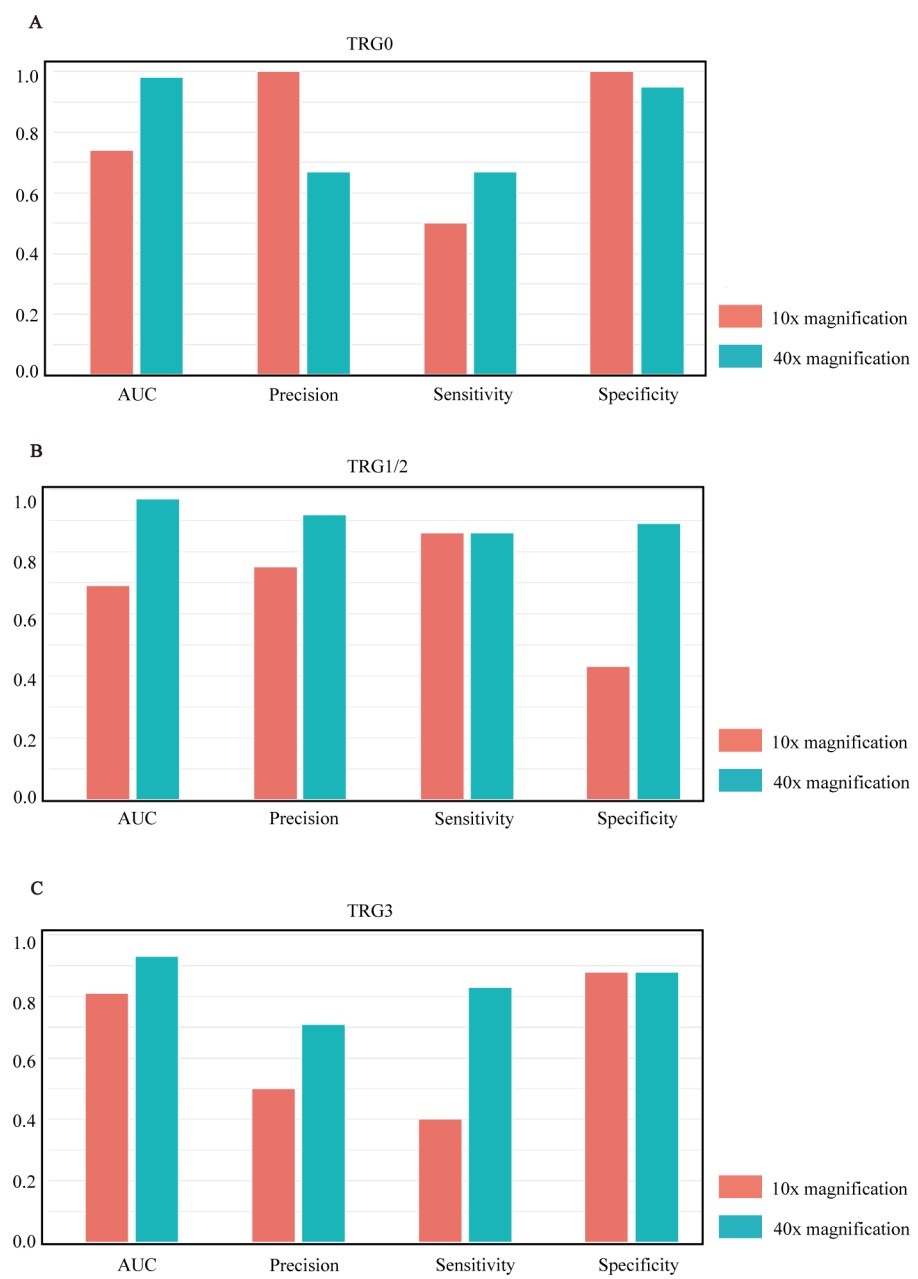

**Figure 3 Model performance histograms of the slides at different magnifications.** (A–C) respectively represent the comparison of AUC, precision, sensitivity, and specificity of the model against TRG0, TRG1/2, and TRG3 at 10× and 40× magnification.

and genomic characteristics to supplement and enhance the predictive ability. There are three studies (*Zhang et al., 2020*; *Shao et al., 2020*; *Feng et al., 2022*) using pathological HE images to predict the curative effect of NAT for rectal cancer. Although these studies have achieved powerful prediction powers, it is only for identification of pCR *vs* non-pCR. These programs, as well as other ML-based programs for radiological images, lack the power of

**Table 4 AUC for three-class classification model at 40× magnification and 10× magnification under the slide level and tile level.**

| category | | At 40× magnification | At 10× magnification |
|---|---|---|---|
| TRG0 | Slide level | 0.98 | 0.74 |
| | Tile level | 0.87 | 0.92 |
| TRG1/2 | Slide level | 0.97 | 0.69 |
| | Tile level | 0.83 | 0.67 |
| TRG3 | Slide level | 0.93 | 0.81 |
| | Tile level | 0.77 | 0.67 |

Notes.
AUC, area under the curve.

**Table 5 The multi-class Matthews' correlation coefficients in the prediction of TRG 0,1/2 and 3 at different magnifications.**

| category | | At 40× magnification | At 10× magnification |
|---|---|---|---|
| TRG0 | Slide level | 0.617 | 0.690 |
| | Tile level | 0.499 | 0.623 |
| TRG1/2 | Slide level | 0.734 | 0.316 |
| | Tile level | 0.501 | 0.233 |
| TRG3 | Slide level | 0.683 | 0.298 |
| | Tile level | 0.425 | 0.221 |

identifying TRG3 patients, the most important group of patients that will not benefit from NAT at all. Therefore, our DL-based algorithm has more clinical significance.

The image of CAM analyses revealed that TILs seem to contribute to the most in our model. TILs are considered to be important anti-tumor cells (*Rosenberg, Spiess & Lafreniere, 1986*). As large numbers of lymphocytes infiltrating into the tumor, they contribute to the response to radiotherapy and chemotherapy. It is known that the number of TILs have positive significance for the tumor response and survival outcome. Accumulation of TILs at tumor sites is taken as a positive prognostic factor in rectal cancer (*Zinovkin et al., 2021*). Therefore, this pathological feature is statistically related to the great response, which provides an explanation for some features extracted by our model when determining the treatment response.

There are limitations of our study: Firstly, data used to build the model is from a single center. Multi-center samples for cross-center validation are needed, in order to enhance the generalization ability of the model. Secondly, the model was based on single scale images. In addition to experiments in the single scale, fusing with 10× and 40× images to jointly build a multi-scale model might improve the performance. Finally, we still need to investigate the relationship between efficacy-related features extracted from pathological images and overall survival outcomes for the disease.

In conclusion, the DL-based multi-class algorithm we established provides a superior prediction power for rectal cancer NAT.

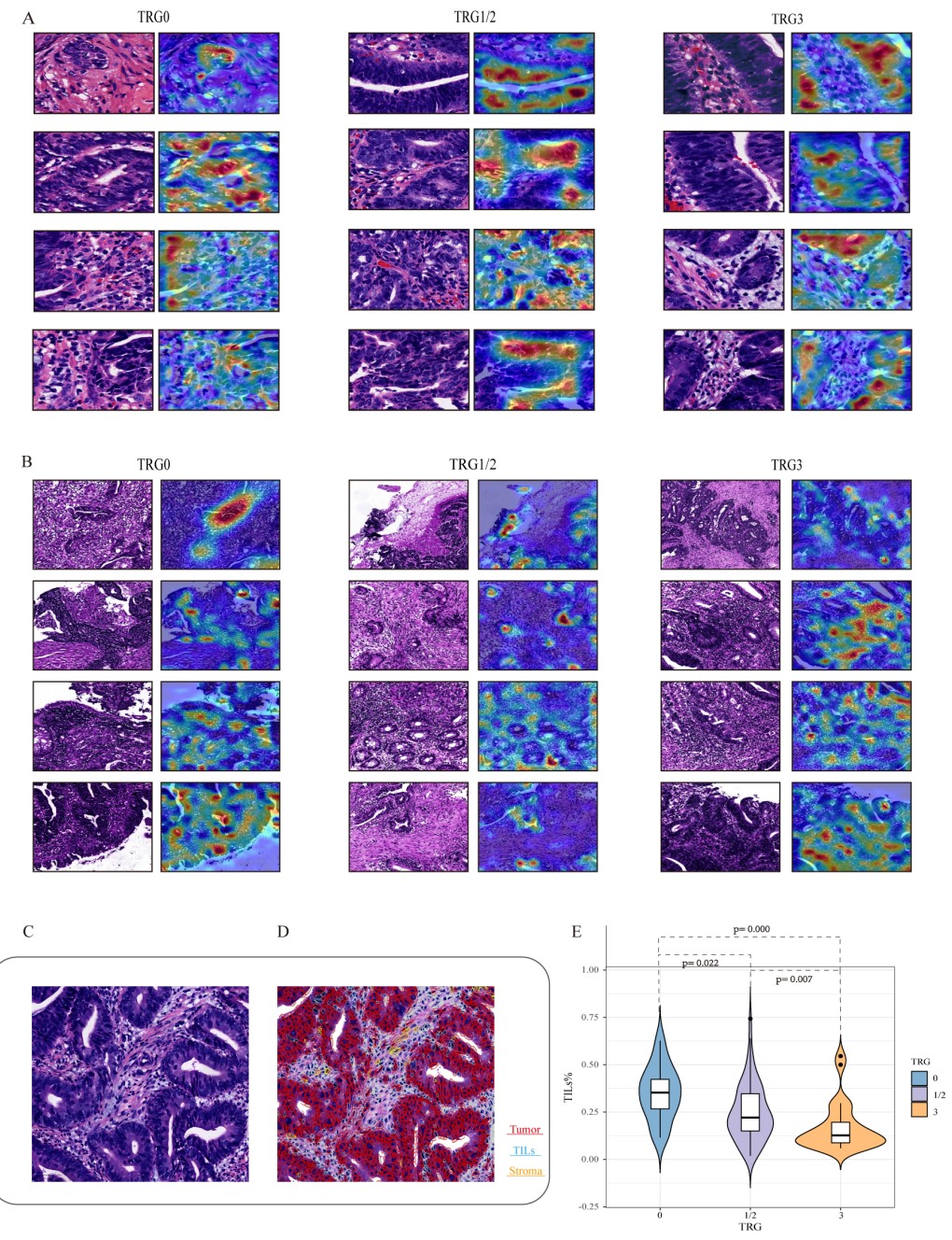

**Figure 4 Visualization of the important regions in HE images at 40× and 10× magnifications for the prediction of NAT.** Based on the CAM algorithm, areas with large contribution from the model are displayed in red, and (A and B) represent the visual expression of different treatment responses at 40× and 10× magnifications. We visualize important local tumor areas in WSIs to explore differences between the efficacy groups. (C and D) represent the representative HE images and the cell quantitative analysis mask. Using the Qupath, segmentation shows red indicates tumor cells, blue indicates TILs, and yellow indicates stromal cells. (E) shows the proportion of TILs was significantly associated with the treatment response.

## ACKNOWLEDGEMENTS

We thank all members of the Xu laboratory for helpful discussions.

### Funding

This work was supported by the National Natural Science Foundation of China [61906022], Chongqing Natural Science Foundation cstc2020jcyj-msxmX0482, Chongqing University Research Fund 2021CDJXKJC004, and Chongqing Medical Scientific Research project (Joint project of Chongqing Health Commission and Science and Technology Bureau) 2020MSXM088. The funders had no role in study design, data collection and analysis, decision to publish, or preparation of the manuscript.

### Grant Disclosures

The following grant information was disclosed by the authors:
The National Natural Science Foundation of China: 61906022.
Chongqing Natural Science Foundation: cstc2020jcyj-msxmX0482.
Chongqing University Research Fund: 2021CDJXKJC004.
Chongqing Medical Scientific Research project (Joint project of Chongqing Health Commission and Science and Technology Bureau): 2020MSXM088.

### Competing Interests

The authors declare there are no competing interests.

### Author Contributions

- Yihan Wu performed the experiments, analyzed the data, prepared figures and/or tables, authored or reviewed drafts of the article, and approved the final draft.
- Xiaohua Liu performed the experiments, prepared figures and/or tables, authored or reviewed drafts of the article, and approved the final draft.
- Fang Liu performed the experiments, authored or reviewed drafts of the article, and approved the final draft.
- Yi Li analyzed the data, authored or reviewed drafts of the article, and approved the final draft.
- Xiaomin Xiong performed the experiments, authored or reviewed drafts of the article, and approved the final draft.
- Hao Sun conceived and designed the experiments, authored or reviewed drafts of the article, and approved the final draft.
- Bo Lin conceived and designed the experiments, authored or reviewed drafts of the article, and approved the final draft.
- Yu Li conceived and designed the experiments, authored or reviewed drafts of the article, and approved the final draft.
- Bo Xu conceived and designed the experiments, authored or reviewed drafts of the article, and approved the final draft.

## Human Ethics

The following information was supplied relating to ethical approvals (i.e., approving body and any reference numbers):

This study was approved by the Chongqing University Cancer Hospital of Medicine Institutional Review Board(Ethical Application Ref: CZLS2022185-A).

## Data Availability

The code and raw measurements are available in the Supplementary Files. The raw data show the probability values of slides and tiles judged to be in each category at different magnifications.

## Supplemental Information

Supplemental information for this article can be found online at http://dx.doi.org/10.7717/peerj.15408#supplemental-information.

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
