# Peer review of "A multi-class classification algorithm based on hematoxylin-eosin staining for neoadjuvant therapy in rectal cancer: a retrospective study"

_PeerJ, doi:10.7717/peerj.15408_

## Round 0.1 · original submission · Major Revisions

All three reviewers give opinions. Please answer the questions carefully, add necessary experimental data, and complete all modifications as far as possible.

Reviewer 1 ·

Basic reporting

The manuscript from Wu and colleagues presents an interesting work on the application of a multi-class classification algorithm for the prediction of NAT treatment response in rectal cancer patients. Overall, the article is clear and well written, and it provides a comprehensive literature in support of the methods. The gap in available screening tools is clearly reported, and the context and importance of the current work highlighted. Supporting figures are appropriate and a great addition to the article, however, I believe the resolution of the images should be improved, as at the current stage it is difficult to read all the details, as they appeared a little grainy.

A small note on Figure 3. I would suggest the authors to move the title of the chart above each plot and not next to the y-axis, as it may be misleading.

For what concerns Table 1, I would suggest the authors to rename the value “neoadjuvant radiotherapy” to something like “underwent neoadjuvant radiotherapy”, as it may be a misleading label, given that undergoing a NAT therapy was an inclusion criteria, but not all of the participants underwent the same treatment.

Finally, I would suggest the authors to check reference 6, as there is an error in the reference list at the end of the manuscript.

Experimental design

The research question of the article is well defined, and the proposed network seems appropriate to test the hypothesis. I have however several concerns regarding the dataset and the approach used to test and validate the results, which I have detailed below.

- My biggest concerns with the research is that multiple tiles were obtained from each participant. Given this, one would expect a K-Fold cross validation in which all the tiles from a single individual, original and augmented are in the same fold. In this situation, we can assume that the model is learning the general features of the image and not learning specific details that can be assigned to a single participant. On the other hand, when features from the same individual are both in the train and test set, we can't exclude that the model is learning the correlation between images, and is assigning a class based on the individual itself, especially given the overlapping between the WSI tiles. It is not clear from the manuscript whether this was done or not. If this is the case, i would suggest the authors to clarify this in the writing (as this seems the case from line 144-155), otherwise I would suggest the authors to run the training and validation of the model ensuring that the tiles of a single participant are in the same fold.

- Continuing on the previous point, the authors indicated that several data augmentation techniques were employed (Line 126). To better understand the impact of the data augmentation procedure, other than ensuring that augmented data are in the same fold as the original data, I would suggest the authors to indicate the amount of augmented tiles used in the dataset, either as in total number or percentage. This is especially important because it is not clear whether the number reported in Line 128-130 refer to the original or augmented dataset.

- A small note, the authors did an amazing job in reporting all the details of the model, and in sharing their code, which is very well written. For reproducibility, I would suggest the authors to also indicate the version of SKLearn on Line 143, given that some of the package methods were employed, as well as of QuPath.

- Similarly, I believe the article could benefit from a short explanation about the hyperparameters tuning, which in the current state of the manuscript is mentioned (Line 148) but not in sufficient details to allow a proper replication.

Validity of the findings

The impact and novelty of the study are clearly reported, and the results and implication properly discussed. The authors did a great job in sharing their code and results, however (unfortunately) the original images are not shared. I assume there is high level of risk in sharing medical imaging data, however if this is the case I believe it must be disclosed in the article

Additional comments

I praise the authors for the quality their work and of their article. The only big critique I have with regard to the methods is the k-fold validation, which as I have detailed above, I believe must be clarified before the article can be considered for publication.

Reviewer 2 ·

Basic reporting

Wu et al. adopted a multi-class deep learning classifier to address the task of tumor regression grade (TRG) prediction in patients with rectal cancer who undergo neoadjuvant therapy. The manuscript is fluent and provides the needed context; the tables are well organized and raw data are available. Figures are of low quality; in particular figure 4 and figure 5 reporting the visualization of the CAM should be bigger.
A more extensive literature review should be included; in particular, the authors claim to propose the first multi-class algorithm for TRG classification on histopathology images (see Jepsen DNM, Høeg H, Thagaard J, Walbech JS, Gögenur I, Fiehn AK. Development of a method for digital assessment of tumor regression grade in patients with rectal cancer following neoadjuvant therapy. J Pathol Inform. 2022 Nov 8;13:100152. doi: 10.1016/j.jpi.2022.100152. PMID: 36605115; PMCID: PMC9808016.)
Also, literature reference for the smooth grad-CAM algorithm is missing.

Experimental design

Although the work does not propose a novel algorithm but adopts a well-known deep learning classifier (ResNet family), the study is relevant for clinical decision in the treatment of locally advanced rectal cancer. In particular, an automated algorithm for TRG classification can be a valuable tool to support clinical histopathology practice.
The authors also compared performance on tiles extracted from WSIs at different magnification level (40x and 10x) to evaluate predictive power of local vs more global histology features for TRG classification. Finally, a visualization of the most relevant features is explored for model interpretability.

However, the experimental pipeline suffers from a potential bias in the splitting design used for the 5-fold cross-validation approach; slides from the same patient are randomly divided into train and test sets (row 146), possibly introducing data leakage and inflating the classification performance (see "Bussola, N., Marcolini, A., Maggio, V., Jurman, G., & Furlanello, C. (2021, January). AI slipping on tiles: Data leakage in digital pathology. In International Conference on Pattern Recognition (pp. 167-182). Springer, Cham.) Thus, experiments should be replicated ensuring that all slides from the same patient are either in the training or in the test set.
Moreover, the reproducibility of the experiments would benefit from more details on the data preprocessing and experimental design:
- line 105: define "poor quality of HE-stained images", did they include out of focus regions or artifacts such as pen marks, tissue folds...?
- lines 121--125: Figure 1 indicates a grid approach for tile cropping, but it is not specified in the main text. Which image was used as reference for the stain normalization? Also, what software/library was used for tile extraction and stain normalization?
- line 129. Is there a specific clinical or computational reason why the authors combine TRG 1 and TRG 2 into a single class?
- line 146. Slides in the CV folds were balanced by TRG class or also by patient metadata (e.g. T-stage or gender)?
- line 168. How many tiles per slide?

Validity of the findings

As commented above, the experimental design must be replicated dividing slides from the same patient in the same partition (train or test sets).

Classification performance are evaluated at tile and slide level in terms of macro- and micro- AUC for different magnification levels. However, it is not clear if results in Table 2 and Figure 2 are computed on the test set; moreover, a 5-folds cross validation procedure is adopted (line 146) thus variability on the folds should be reported on the training/test sets.

The visual analysis of relevant features for the CNN classifier via the CAM algorithm is an useful approach for model interpretability; further details are needed on how the presence of tumor-infiltrating lymphocytes was evaluated on HE tiles to validate the findings.

Additional comments:
- please specify the Python library used to compute ROC/AUC
-Given the unbalanced nature of the dataset (63% of the slides are from the TGR1/2 class), I would also suggest to include the multi-class Matthews' correlation coefficients to evaluate classification performance.

Additional comments

Please correct the following:
- line 118: replace "were" with "was"
- line 148: "Finally, the optimal hyperparameters of the model were confirmed." sentence not clear
- line 184: Incomplete sentence
- line 192: CAM is a post-hoc explanation of a trained model, not an interpretable model, please reformulate.
- line 212: remove "superb"
- line 213: remove "in"
- caption of Table 2 and Figure 5: add details

Reviewer 3 ·

Basic reporting

This article explores a framework for neoadjuvant therapy in rectal cancer based on hematoxylin-eosin staining. The authors now propose a scheme that aims to classify three groups (TRG0, TRG1/2 and TRG3) using a neural network (ResNet). Tumor assessments were performed by two pathologists, one responsible for delineation and the other for examination. The author creates a dataset from 74 patients. 113 H&E stained slides used for experiments.

This paper is well organized in content.

Experimental design

The following sections of the experimental part are not fully understood.
• How was the magnification scale (x10 and x40) determined?
• A patient slide can be in both training and test groups. This may cause the test results to be high. The same is true for data augmentation. Was this taken into account during 5-fold cross-validation?

Validity of the findings

When the deficiencies in the experimental part are corrected, there will be no deficiency in the findings.

Additional comments

This article explores a framework for neoadjuvant therapy in rectal cancer based on hematoxylin-eosin staining. The authors now propose a scheme that aims to classify three groups (TRG0, TRG1/2 and TRG3) using a neural network (ResNet). Tumor assessments were performed by two pathologists, one responsible for delineation and the other for examination. The author creates a dataset from 74 patients. 113 H&E stained slides used for experiments

• How was the magnification scale (x10 and x40) determined?
• A patient slide can be in both training and test groups. This may cause the test results to be high. The same is true for data augmentation. Was this taken into account during 5-fold cross-validation?

Although this article is well organized in terms of content, it may be useful to eliminate the shortcomings mentioned above.

---

## Round 0.2 · accepted · Accept

There are three reviewers in total. One thinks it is acceptable, one thinks it needs minor revision, and the last one thinks it needs major revision, but they all declined to review it further. On the whole, this study has no obvious risk of publication and basically meets the requirements of publication.

Reviewer 1 ·

Basic reporting

'no comment'

Experimental design

'no comment'

Validity of the findings

'no comment'

Additional comments

I am grateful to the authors for taking mine and the other reviewers comments into consideration. I believe the manuscript has greatly improved from its previous iterations, and the most critical aspects of the article has been addressed. The article is overall more clear now, and methods reported with a sufficient level of details to allow for replication. Figures are now of good quality.

I would suggest the editor to accept the manuscript for publication in its current form.